# Fiber-like Action of d-Fagomine on the Gut Microbiota and Body Weight of Healthy Rats

**DOI:** 10.3390/nu14214656

**Published:** 2022-11-03

**Authors:** Sara Ramos-Romero, Julia Ponomarenko, Susana Amézqueta, Mercè Hereu, Bernat Miralles-Pérez, Marta Romeu, Lucía Méndez, Isabel Medina, Josep Lluís Torres

**Affiliations:** 1Institute for Advanced Chemistry of Catalonia (IQAC-CSIC), E-08034 Barcelona, Spain; 2Department of Cell Biology, Physiology & Immunology, Faculty of Biology, University of Barcelona, E-08028 Barcelona, Spain; 3Centre for Genomic Regulation (CRG), The Barcelona Institute of Science and Technology, Universitat Pompeu Fabra (UPF), E-08003 Barcelona, Spain; 4Departament d’Enginyeria Química i Química Analítica and Institut de Biomedicina (IBUB), Universitat de Barcelona, E-08028 Barcelona, Spain; 5Facultat de Medicina i Ciències de la Salut, Universitat Rovira i Virgili, E-43201 Reus, Spain; 6Instituto de Investigaciones Marinas (IIM-CSIC), E-36208 Vigo, Spain

**Keywords:** body weight, gut microbiota, inflammation, d-fagomine, iminosugar, iminocyclitol

## Abstract

The goal of this work is to explore if the changes induced by d-fagomine in the gut microbiota are compatible with its effect on body weight and inflammation markers in rats. Methods: Sprague Dawley rats were fed a standard diet supplemented with d-fagomine (or not, for comparison) for 6 months. The variables measured were body weight, plasma mediators of inflammation (hydroxyeicosatetraenoic acids, leukotriene B4, and IL-6), and the concentration of acetic acid in feces and plasma. The composition and diversities of microbiota in cecal content and feces were estimated using 16S rRNA metabarcoding and high-throughput sequencing. We found that after just 6 weeks of intake d-fagomine significantly reduced body weight gain, increased the plasma acetate concentration, and reduced the plasma concentration of the pro-inflammatory biomarkers’ leukotriene B4, interleukin 6 and 12 hydroxyeicosatetraenoic acids. These changes were associated with a significantly increased prevalence of *Bacteroides* and *Prevotella* feces and increased *Bacteroides*, *Prevotella*, *Clostridium,* and *Dysgonomonas* while reducing *Anaerofilum*, *Blautia,* and *Oribacterium* in cecal content. In conclusion, d-fagomine induced changes in the composition and diversity of gut microbiota similar to those elicited by dietary fiber and compatible with its anti-inflammatory and body-weight-reducing effects.

## 1. Introduction

d-Fagomine, ((2R, 3R, 4R)-2-hydroxymethylpiperidine-3,4-diol; 1,2-dideoxynojirimycin), is a polyhydroxylated piperidine (a saturated six-atom ring formed of five carbon atoms and a nitrogen atom). The spatial configurations of the hydroxyl groups in d-fagomine coincide with those of simple sugars such as d-glucose and d-mannose.

d-Fagomine was first isolated from seeds of buckwheat (*Fagopyrum esculentum* Moench, Polygonaceae) [1] and later found in other plant sources such as the leaves of *Morus bombycis* (Moraceae); root bark, fruit, and leaves of mulberry (*Morus alba*, Moraceae); roots of *Lycium chinense* (Solanaceae); roots and leaves of *Xanthocercis zambesiaca* (Fabaceae); pods of *Angylocalyx pynaertii* (Fabaceae); and leaves of *Baphia nitida* (Fabaceae) [2,3]. As d-fagomine is chemically and biologically stable, it remains unaltered during boiling, baking, frying, and fermentation [4]. d-fagomine has been part of the diet in European, Asian, and American countries, mainly as a component of buckwheat-based foodstuffs, such as noodles, groats, pancakes, boiled flour, fried dough, beer, cookies, and bread. The estimated total intake of d-fagomine resulting from a diet that includes such foodstuffs as the only source of carbohydrates is between 3 and 17 mg per day [4].

As an intestinal glycosidase inhibitor, d-fagomine reduces the postprandial blood glucose concentration after oral administration of either sucrose or starch, as evidenced in both rats [5] and humans (results accessible at https://clinicaltrials.gov/ct2/show/NCT01811303). Other observations have revealed a second possible action of d-fagomine and maybe also of other iminocyclitols: the selective inhibition of bacterial adhesion to the intestinal mucosa. In vitro studies have shown that d-fagomine selectively agglutinates fimbriated Enterobacteriales, such as *Escherichia coli* and *Salmonella enterica* serovar Typhimurium, and consequently, it inhibits the adhesion of these bacteria to pig intestinal mucosa [5]. These results suggest that d-fagomine may modify the composition of the gut microbiota. The first evidence supporting this hypothesis was the observation that d-fagomine counteracted diet-induced increases in the populations of gut Enterobacteriales while reducing weight gain [6]. We later suggested that the action of d-fagomine on some microbial populations might explain, at least in part, its functional effect against fat-induced impaired glucose tolerance and hepatic inflammation [7]. d-Fagomine has the capacity to stabilize levels of putatively beneficial gut bacteria (e.g., the Bacteroidetes phylum, Bifidobacteriales order, and *Prevotella* genus) in normal rats fed a standard diet for an extended period of time [8,9]. When combined with fish omega-3 polyunsaturated fatty acids (PUFAs), d-fagomine has an early effect compatible with fast changes in bacterial composition, while the protective anti-inflammatory action of PUFAs appears to be less related to changes in gut microbiota [9].

As evidence of the iminocyclitol d-fagomine having a decisive influence on the composition of gut microbiota is accumulating, we have embarked on a thorough characterization of these changes. We present here a study of the microbial composition in the gut (feces and cecal content) of Sprague Dawley rats administered d-fagomine for a period of six months and compare it with gut microbial composition in control animals. We also examine the possible association of the observed changes in microbial populations with the effects of d-fagomine on body weight and inflammation markers.

## 2. Materials and Methods

### 2.1. Experimental Design and Sample Collection

A total of 18 male Sprague Dawley rats from Envigo (Indianapolis, IN, USA), aged 8–9 weeks, were used. All animal handling was conducted in the morning to minimize the effects of circadian rhythms. The rats were housed (*n* = 3 per cage) under controlled conditions of humidity (60%), and temperature (22 ± 2 °C) with a 12 h light-12 h dark cycle. They were randomly divided into 2 groups (*n* = 9/group): the control group was fed a standard diet of 2014 Teklad Global 14% Protein chow from Envigo (STD); and the group was fed the same standard diet supplemented with 0.96 g d-fagomine/kg feed (FG). The dose of d-fagomine was established based on the results of previous studies of bacterial adhesion in vitro [5] and of its in vivo effects on gut Enterobacteriales [6]. The animals were fed *ad libitum* with free access to feed and water (Ribes, Barcelona, Spain).

Feed consumption and body weight were monitored three times per week throughout the experiment. Energy intake was calculated as an estimate of metabolizable energy based on the Atwater factors, assigning 4 kcal/g protein, 9 kcal/g fat, and 4 kcal/g available carbohydrates. Fecal samples were collected by abdominal massage after week 23.

An oral glucose tolerance test (OGTT) was performed on fasted animals. A solution of glucose (1 g/kg body weight) was administered by oral gavage before the test, and blood glucose concentration was measured 15, 30, 45, 60, 90, and 120 min after glucose intake. Blood glucose concentration was measured by the enzyme electrode method, using an Ascensia ELITE XL blood glucose meter (Bayer Consumer Care, Basel, Switzerland).

After 6 months of supplementation, the rats were fasted overnight and anesthetized intraperitoneally with ketamine from Merial Laboratorios (Barcelona, Spain) and xylacine from Quimica Farmaceutica (Barcelona, Spain) (80 and 10 mg/kg body weight, respectively). Blood samples were collected intracardiac puncture, and plasma was separated by centrifugation. Cecal contents were extracted, weighed, and immediately frozen in liquid N_2_. All the samples were stored at −80 °C until analysis.

### 2.2. Plasma Aminotransferases, Insulin, and Cholesterol

Plasma aspartate aminotransferase (AST) and alanine aminotransferase (ALT) activities were measured by means of spectrophotometry using the corresponding commercial kits (Spinreact, Girona, Spain) in a COBAS MIRA autoanalyzer (Roche Diagnostics System, Madrid, Spain).

Plasma insulin levels were measured in fasted animals using the rat/mouse insulin enzyme-linked immunosorbent assay (ELISA) kit from Millipore Corporation (Billerica, MA, USA).

Plasma total cholesterol, high-density lipoprotein (HDL) cholesterol, and low-density lipoprotein (LDL) cholesterol were measured using a spectrophotometric method and the corresponding kits from Spinreact (Girona, Spain).

### 2.3. DNA Extraction and Sequencing

Total DNA was extracted individually using a QIAampTM DNA Stool Mini Kit from QIAGEN (Hilden, Germany) from both feces and cecal content and quantified using a Nanodrop 8000 Spectrophotometer (ThermoScientific, Waltham, MA, USA). All DNA samples were diluted to 5 ng/µL and used to amplify the V3–V4 regions of the 16S ribosomal RNA (rRNA) gene, using the following universal primers in a limited cycle PCR:
forward primer, 5′ TCG GCA GCG TCA GAT GTG TAT AAG AGA CAG CCT ACG GGN GGC WGC AG;reverse primer, 5′ GTC TCG TGG GCT CGG AGA TGT GTA TAA GAG ACA GGA CTA CHV GGG TAT CTA ATC C.

The V3–V4 region of the 16S rRNA gene was sequenced on an Illumina MiSeq platform at the Servei de Genòmica i Bioinformàtica of the Universitat Autònoma de Barcelona (Bellaterra, Cerdanyola del Vallés, Spain) following the manufacturer’s guidelines and instructions.

The 16S rRNA OTU counts were analyzed using the R package of Phyloseq (version 1.30.0) [10]. Statistical significance of the relative abundances of taxa was evaluated using the Wald test in the DESeq2 package (version 1.30.1) [11].

### 2.4. Acetic Acid Concentration

Acetic acid (the major short-chain fatty acid, SCFA) was analyzed in plasma and fecal samples after 6 months of supplementation by gas chromatography using a previously described method [12] with some modifications. Briefly, the freeze-dried feces (~50 mg dry matter) or plasma (~50 µL) and a solution (1.5 mL) containing the internal standard 2-ethylbutyric acid (6.67 mg/L) and oxalic acid (2.97 g/L) in acetonitrile/water (3:7 for feces and 7:3 for plasma) was added. Then, SCFAs were extracted for 10 min using a rotating mixer. The suspension was centrifuged, and the supernatant was filtered through a 0.45 µm nylon filter. Next, an aliquot of the supernatant (0.7 mL) was diluted with acetonitrile/water 3:7 to a final volume of 1 mL. After this, the SCFAs were analyzed using a Trace2000 gas chromatograph coupled to a flame ionization detector (ThermoFinnigan, Waltham, MA, USA) equipped with an INNOWax capillary column, 30 m × 530 µm, with a 1 µm film unit (Agilent, Santa Clara, CA, USA). The Chrom-Card software ver. 2.4.1. was used for data processing. Helium was used as the carrier gas with a linear velocity of 5 mL/min. The GC oven temperature was programmed as follows: 80 °C (hold for 1 min) to 120 °C at 15 °C/min (hold for 4 min) to 130 °C at 5 °C/min (hold for 4 min) to 235 °C at 8 °C/min (hold for 4 min). A flame ionization detector (FID) was used at a base temperature of 240 °C.

### 2.5. Plasma Mediators of Inflammation

Several hydroxyeicosatetraenoic acids (HETEs) and leukotriene B4 (LTB4), lipid mediators derived from the metabolism of arachidonic acid (ARA), were determined in plasma by liquid chromatography coupled with tandem mass spectrometry (LC-MS/MS), using a method modified from Dasilva et al. [13]. Erythrocyte-free plasma samples (90 µL) were thawed, diluted in the presence of butylated hydroxytoluene (BHT), and spiked with the internal standard (12HETE-d8; Cayman Chemicals, Ann Arbor, MI, USA). Then, the samples were centrifuged (800× *g*, 10 min), and the lipids in the supernatants were purified by solid-phase extraction (SPE). The LC-MS/MS analyzer consisted of a Dionex UltiMate 3000 Series chromatograph (Thermo Fisher, Rockford, IL, USA) coupled to a dual-pressure linear ion-trap mass spectrometer, LTQ Velos Pro (Thermo Fisher, Rockford, IL, USA) operating in negative electrospray ionization (ESI) mode. A Symmetry C18 Column 3.5 µm, 150 mm × 2.1 mm inner diameter (Waters, Milford, MA, USA) with a C18 4 × 2 mm guard cartridge (Phenomenex, Torrance, CA, USA) was used in the separation step. Samples (10 µL) were eluted with a binary system consisting of 0.02% aqueous formic acid [A] and 0.02% formic acid in methanol [B] under gradient conditions of: 0 min, 60% B; 2 min, 60% B; 12 min, 80% B; 13 min, 80% B; 23 min, 100% B; 25 min, 100% B; and 30 min, 60% B, at a flow rate of 0.2 mL/min.

Concentrations of plasma IL-6 were determined using Milliplex xMAP multiplex technology on a Luminex xMAP instrument (Millipore, Austin, TX, USA). The MILLIPLEX Analyst 5.1 (VigeneTech; Carlisle, PA, USA) software was used for data analysis.

### 2.6. Statistical Analysis

All data manipulation and statistical analyses were performed using GraphPad Prism 5 (GraphPad Software, San Diego, CA, USA). Statistical significance was determined by two-way ANOVA for repeated measures of body weight or the Wilcoxon or Student’s *t* test to compare the groups studied. PERMANOVA was applied for each β-diversity determination. The results are expressed as means with their standard errors (SEM). Differences were considered significant when *p* < 0.05.

## 3. Results

### 3.1. Body and Cecum Weights, Feed Intake, and OGTT

The animals in the FG group gained less weight than those in the STD group from week 6 until the end of the experiment (Figure 1A), while there were no significant differences in the OGTT (Figure 1B).

Feed intake (STD: 4.6 g/day/100 g body weight, SEM 0.5; FG: 4.8 g/day/100 g body weight, SEM 0.5) and energy intake (STD: 13.3 kcal/day/100 g body weight, SEM 1.4; FG: 13.8 kcal/day/100 g body weight, SEM 1.2) were similar for both groups throughout the experiment.

At the end of the study, there were no differences in cecum weight between the groups (STD: 5.1 g, SEM 0.4; FG: 5.2 g, SEM 0.2).

### 3.2. Plasma Aminotransferases, Insulin, and Cholesterol

The animals in the FG group did not show any difference in plasma aminotransferases, fasting insulin, or cholesterol levels at the end of the study (Table 1).

### 3.3. Relative Composition of Microbial Communities

There were no differences between the groups at the phylum taxonomical level. In feces, the class Bacteroidia was significantly (*p* < 0.05) increased by d-fagomine supplementation, while the class Flavobacteriia was reduced. Similar changes were detected in the orders Bacteroidales and Flavobacteriales. There were no changes in cecal content at these taxonomical levels.

At the family level, d-fagomine intake significantly reduced Flavobacteriaceae and Odoribacteraceae while increasing Bacteroidaceae and Prevotellaceae in feces (Figure 2A). These last two families, as well as Peptostreptococcaceae, were also increased in cecal content (Figure 2C), while Lachnospiraceae was significantly (*p* < 0.05) reduced.

At the genus level, d-fagomine supplementation significantly (*p* < 0.05) increased *Bacteroides* and *Prevotella* while reducing *Flavobacterium* and *Odoribacter* in feces (Figure 2B). The increased genera in cecal content were also *Bacteroides* and *Prevotella*, together with *Clostridium* and *Dysgonomonas* (Figure 2D), while the genera *Anaerofilum*, *Blautia* and *Oribacterium* were significantly reduced by d-fagomine intake.

### 3.4. α- and β-Diversities of the Fecal and Cecal Microbiota

The α-diversity was similar in the two groups at the phylum level. The Shannon index indicated a reduction (*p* < 0.05 at the class and order levels, *p* < 0.01 at the family and genus levels; Figure 3A,B) of the α-diversity in fecal samples from the FG group, while α-diversity was increased in cecal samples (*p* < 0.05 at the family level; Figure 3A). Similar results were obtained using the inverse Simpson index (Figure 3C,D).

In fecal samples, β-diversity was similar at the phylum, class, and order levels (Table 2). The weighted UniFrac distance was significantly (*p* < 0.05) different at the family, genus, and species levels between controls and animals supplemented with d-fagomine. In cecal content, the weighted UniFrac distance was significantly (*p* < 0.05) different between the STD and FG groups at all the taxonomical levels.

### 3.5. Plasma and Fecal Acetic Acid

d-Fagomine supplementation (FG group) increased the plasma acetate content with respect to the STD group (*p* < 0.05, Figure 4A), while the differences were not significant in feces (Figure 4B).

### 3.6. Biomarkers of Inflammation

Plasma LTB4 and IL-6 were significantly (*p* < 0.05) lower in animals after 6 months of supplementation with d-fagomine with respect to those not supplemented (Figure 5).

Furthermore, the plasma concentration of 12HETE was significantly (*p* < 0.05) reduced in supplemented animals (FG group), while the other ARA-derived pro-inflammatory hydroxyeicosatetraenoic acids were not modified by d-fagomine supplementation (Table 3).

## 4. Discussion

In agreement with our previous observations [9], d-fagomine supplementation had a reducing effect on body weight gain in healthy rats after just 1.5 months (Figure 1A). Previous studies also showed that d-fagomine reduced body weight gain in rats fed energy-dense diets [6,7,14]. Therefore, body weight control is a consistent effect of d-fagomine in both normoweight and overweight animals. To explain the reduction in body weight, we first focused on gut Enterobacteriales, as this group had previously been associated with weight gain and inflammation [6,15]. We later found that d-fagomine had a more pronounced influence on other microbial orders, such as Prevotellaceae, Lactobacilliaceae, and Bifidobacteriaceae [8,9]. So, to better assess the changes induced by d-fagomine at various taxonomical levels, here we address the study of the gut genome by 16S rRNA gene sequencing techniques.

Our results from the present study show that d-fagomine induces changes in gut microbiota at different taxonomical levels. Some changes are consistent at the levels of class (increased populations of Bacteroidia, reduced populations of Flavobacteriia), order (increased Bacteroidales, reduced Flavobacteriales), family (increased Bacteroidaceae and Prevotellaceae, reduced Flavobacteriaceae) and genus (increased *Bacteroides* and *Prevotella*, reduced *Flavobacterium* and *Odoribacter*) in feces. At the genus level, *Bacteroides* and *Prevotella* also increased in cecal content (Figure 2D). This result is particularly interesting because cecal content might be a more faithful indicator than feces of effective interactions with the intestinal mucosa. As collecting feces is much easier than obtaining samples from intestinal mucus or epithelial tissue, most of the relations between microbiota and health in humans have been established by studying fecal populations and physiologically relevant variables [16]. While the examination of fecal microbiota has evident diagnostic and clinical significance, cecal content might provide more physiologically relevant information. Significant differences in microbial populations within the mucosal and luminal niches have been reported in healthy humans [17]. In mice, fecal and cecal microbiota profiles are also clustered in different ways, so the fecal microbiome is not necessarily representative of the population actually in contact with the mucosa along the whole of the intestinal tract [18,19]. In our study, the fact that increased populations in both feces and cecal content were only observed for *Bacteroides* and *Prevotella* indicates that the most dramatic effect of d-fagomine within the rat intestine may involve these two genera (Figure 2B,D). *Bacteroides* spp. is microorganisms that are specialized in degrading dietary saccharides such as short-chain fructooligosaccharides (FOS), arabinoxylans, galactoligosacharides, and xylooligosaccharides, as well as resistant starch [20,21]. *Prevotella* spp., particularly the *Prevotella copri* complex, also exhibits saccharolytic activity, particularly against xyloglucan [22]. The genus *Prevotella* is increasingly appearing as a key mediator of the relationship between nutrition, human metabolism, and health [23]. Intriguingly, as increased *Prevotella* populations have been observed in individuals under conditions of both health and disease, their role in human metabolism is still controversial [23]. Whatever that might be, there is overwhelming evidence that *Prevotella* spp. in oral and intestinal microbiomes is more abundant in high-fiber-consuming rural populations than in high-protein and high-fat-consuming westernized populations [24,25]. Even at the species level, westernization of the diet leads to a loss of diversity, as shown by the underrepresentation of clades of the *P. copri* complex [26]. Since *P. copri* has been mechanistically linked to an improvement in glucose tolerance in mice [27], it may participate decisively in the protective effect of dietary fiber on glucose metabolism. In contradistinction, other results, also in mice, assign an effect of triggering insulin resistance to *P. copri* via the biosynthesis of branched-chain amino acids [28]. Previous results from our group showed that increased populations of *Prevotella* associated with the intake of d-fagomine did not have any significant effect on the fasting blood levels of either glucose or insulin in healthy rats [9]; indeed, such populations even appeared together with a reduction in the elevation of the Homeostatic Assessment Model for Insulin Resistance (HOMA-IR) driven by a high-fat diet in rats [29]. All these apparently contradictory results, as well as those reported by many other authors, highlight the importance of addressing the study of the whole intestinal ecosystem in relation to the host’s metabolism. A given microorganism is not intrinsically beneficial or detrimental by itself, and it can play different roles depending on its relationship with other microbial populations. So, it is important to look at other minor variations and the overall effects on the diversity of the microbial ecosystem.

Apart from the changes in Prevotellaceae, d-fagomine induced other statistically significant changes in the bacterial populations at both the family and genus levels. At the family level, Flavobacteriaceae and Odoribacteraceae were reduced in feces (Figure 2A), while Peptostreptococcaceae were increased and Lachnospiraceae reduced in cecal content (Figure 2C). These differences in population shifts between feces and cecal content emphasize the need to consider fecal populations as markers but not necessarily as being responsible for biological effects at the intestinal or systemic level. Increased populations of Flavobacteriaceae are consistent with the reduced intracellular levels of reactive oxygen species (ROS) detected in the worm *Caenorhabditis elegans* when treated with a prebiotic polysaccharide from the microalga *Spirulina* (Arthrospira) *platensis* {Chen, 2020 #16}. At the genus level, *Clostridium* and *Dysgonomonas* were increased by d-fagomine supplementation, while *Anaerofilum*, *Blautia,* and *Oribacterium* were significantly reduced in the cecal content (Figure 2D). The strongest reduction was detected in *Blautia*: a genus belonging to the Lachnospiraceae family. Reports on the effects of *Blautia* are also contradictory. As relative fecal and cecal abundances of *Blautia* have been related to increased serum concentrations of glucose and insulin in pigs [30], low levels of this genus may be associated with beneficial effects. In support of this possibility, elevated populations of *Blautia* have been recorded in mice fed a high-fat diet showing impaired glucose tolerance [31]. In humans, fecal *Blautia* has been inversely associated with visceral fat area [32]. In contrast, high *Blautia* populations have been associated with disturbances of glucose metabolism and arterial hypertension, and they strongly correlate with low consumption of resistant starch [33]. The influence of soluble prebiotic dietary fiber on the populations of *Blautia* and the whole Lachnospiraceae family is also controversial. Lachnospiraceae is abundant in the cecum of mice given a prebiotic oligosaccharide from lotus seed [34]. In contrast, inulin (3 g/kg/day) and *Lycium barbarum* polysaccharides (400 mg/kg) reduced *Blautia* and *Desulfovibrio* while increasing *Bifidobacterium*, *Lactobacillus,* and *Alistipes* in diabetic Sprague-Dawley rats fed a high-fat diet and injected with streptozotocin (STZ) [35]. With some exceptions, the overall picture emerging from most of the literature suggests that good dietary habits relate to low levels of *Blautia* in both human and animal models.

A reduction in the populations of the genus *Oribacterium* in the animals given d-fagomine contributed to the reduction recorded for the Lachnospiraceae family (Figure 2) in the cecum. Very little is known about the role of *Oribacterium* in the intestinal ecosystem. Our results suggest that closer attention should be paid to the contribution of this genus to the effects of Lachnospiraceae.

d-Fagomine also triggers a small increase in the genus *Dysgonomonas* (Figure 2D). *Dysgonomonas* forms a phylogenetic cluster within the *Bacteroides*-*Prevotella*-*Porphyromonas* group [36]. Again, there is very limited information on the effects of intestinal *Dysgonomonas* on the host physiology apart from the observation that it may be involved in SCFA generation in rats [37]. In vitro fermentation studies suggest that the proliferation of *Dysgonomonas* is fostered by prebiotics such as alginate (arabinoxylans) oligosaccharides [38]. *Dysgonomonas* may reveal itself to be an important player in the maintenance of balanced gut microbiota as it expresses esterases that facilitate the utilization of prebiotics such as xylan polysaccharides by removing acetyl and feruloyl groups [39].

### 4.1. Systemic Acetate and Inflammatory Markers

The strong effect of d-fagomine on the populations of fiber-processing *Prevotella* and *Bacteroides* agrees with the increased blood acetate levels observed in our study (Figure 5). Acetate is the most abundant bacterial metabolite both in the intestinal tract and systemically, in humans and also in animal models [40,41], and it influences body weight and insulin sensitivity in rats [42]. *Prevotella* is emerging as a key player in the conversion of soluble dietary fiber into SCFAs, particularly acetate and butyrate [43]. A fiber-like effect of d-fagomine is consistent with the increased blood acetate concentration and the effect of lowering body weight in Sprague Dawley rats reported here (Figure 4 and Figure 1A, respectively). This is also consistent with the lowering of the biomarkers of low-grade inflammation (IL-6, leucotriene LTB4, and hydroxyeicosatetraenoic acid 12-HETE) summarized in Figure 5 and Table 3. Reduced blood levels of IL-6 have been associated with the intake of dietary fiber in humans [44] and rats [45]. LTB4 is a chemotactic lipid mediator of inflammation generated from arachidonic acid (ARA) during the early phase of inflammation [46]. HETEs are also early mediators of inflammation derived along the ARA pathway [46]. The differential effect of d-fagomine on hydroxyeicosatetraenoic acids (Table 3) may help to elucidate the physiological role of 12-HETE, which remains unclear [47]. Variations in other genera, such as *Blautia* and *Dysgonomonas,* might contribute to these changes. Our results concerning inflammatory mediators show that the action of d-fagomine includes the prevention of age-related low-level systemic inflammation, which is consistent with its fiber-like effect on the intestinal microbiota. The results are also consistent with the effect of d-fagomine on body weight gain, as fat accumulation has been related to the low-level inflammation triggered by diet- and aging-related moderate dysbiosis [48].

### 4.2. Diversity

d-Fagomine increased α-diversity (Shannon and inverse Simpson indexes, Figure 3) at the family level in the cecum while it reduced diversity in feces. As microbial α-diversity is positively associated with fiber intake and the metabolic health of the host [49], the effect of d-fagomine, which lowers weight gain, may, at least in part, be attributed to the promotion of diversity in the cecum. The differences in cecal contents between treated and non-treated rats were not significant at the genus level (Figure 3B,D). The reduced diversity in feces after d-fagomine intake may reflect the threefold increase in Prevotellaceae and particularly *Prevotella* (Figure 2A,B). It should be noted that both the Shannon and inverse Simpson indexes measure not only the total number of taxonomic units but also the relative size of their populations (evenness) [50]. In the cecum, the increase in Prevotellaceae is lower and combined with a reduction in Lachnospiraceae and in its constituting genera *Blautia* and *Oribacterium* (Figure 2C,D). The overall result is increased α-diversity. Then the reduction in α-diversity in feces may be the direct consequence of the physiological regulation of intestinal eubiosys as a response to an elevated increase in Prevotellaceae, resulting in an elevated excretion of microorganisms pertaining to this taxon.

The comparative variation of species between two ecological niches or the variation between two-time points within the same zone is known as β-diversity. In our study, β-diversity, as measured by the weighted UniFrac distance (containing information on both phylogenetic distance and abundance), was different at the family, genus, and species levels between controls and the animals supplemented with d-fagomine in both feces and cecal content (Table 2). This means that the changes induced by d-fagomine supplementation in particular subgroups of microbial populations discussed above translate to significant variations when considering the whole ecosystem. The increase in α- and β-diversities induced in the cecum indicates that d-fagomine may help to reinstate the intestinal microbiota of city-dwelling populations feeding on fiber-poor westernized diets to a more diverse condition associated with rural life. Since it is supplemented in a proportion as low as 0.1%, d-fagomine might help to bolster the benefits of dietary fiber by promoting the proliferation of fiber-processing bacteria. These fiber-like effects of d-fagomine are evidenced at doses around 100-fold lower than those used for soluble dietary fiber (e.g., fructo oligosaccharides, inulin) in rats and mice [51]. This suggests that d-fagomine intake might result in a reduced percentage of ingested dietary fiber, still producing beneficial effects on gut microbiota, particularly an increase in *Prevotella* and *Bacteroides* spp. Therefore, d-fagomine might help to make healthy food more palatable by reinforcing the action of dietary fiber.

## 5. Conclusions

In conclusion, our results show that d-fagomine exerts a fiber-like effect on the gut microbiota of healthy rats by increasing the populations of Prevotellaceae (mainly *Prevotella*) and Bacteroidaceae (mainly *Bacteroides*) as well as by reducing Lachnospiraceae (*Blautia* and *Oribacterium*). This fiber-like effect is compatible with increased levels of blood acetate, reduced levels of age-related inflammatory markers, and reduced body weight gain.

## Figures and Tables

**Figure 1 nutrients-14-04656-f001:**
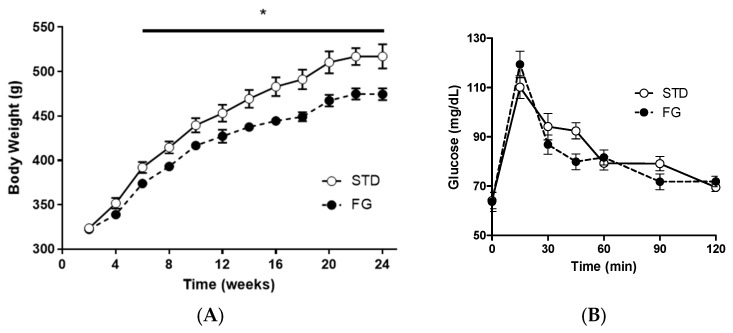
(**A**) Body weight evolution and (**B**) OGTT of rats (*n* = 8–9) supplemented or not with d-fagomine for 6 months. Comparisons were conducted using ANOVA for repeated measures. * *p* < 0.05.

**Figure 2 nutrients-14-04656-f002:**
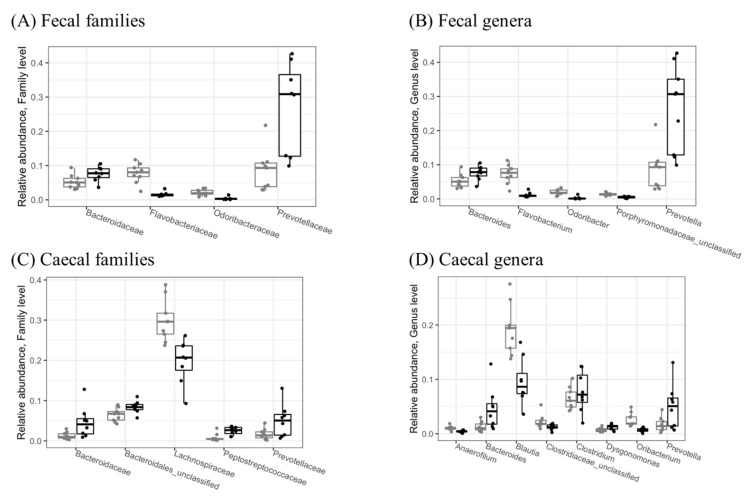
Relative abundances of significantly different (*p* < 0.05) microbial families (**A**,**C**) and genera (**B**,**D**) in fecal (**A**,**B**) and cecal (**C**,**D**) samples collected from the STD (grey) and FG (black) groups of rats (*n* = 8–9) at the end of the study (week 23). Only taxa with relative abundance > 1% are shown. Statistical significance was evaluated using the Wald test in DESeq2, version 1.30.1.

**Figure 3 nutrients-14-04656-f003:**
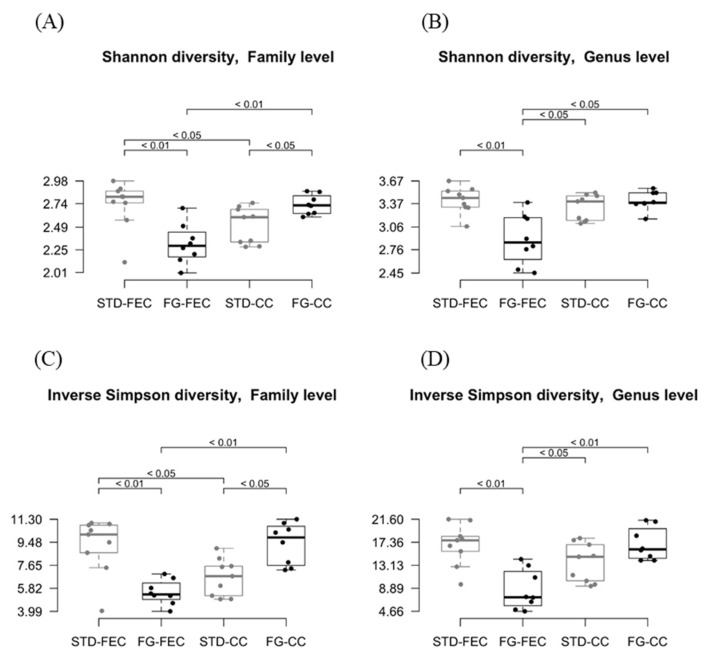
Estimates of Shannon (**A**,**B**) and inverse Simpson (**C**,**D**) α-diversities of fecal and cecal microbiota at family (**A**,**C**) and genus (**B**,**D**) levels in the STD (grey) and FG (black) groups of rats (*n* = 8–9/group) at the end of the study (week 23). Statistical significance was assessed using the Wilcox test.

**Figure 4 nutrients-14-04656-f004:**
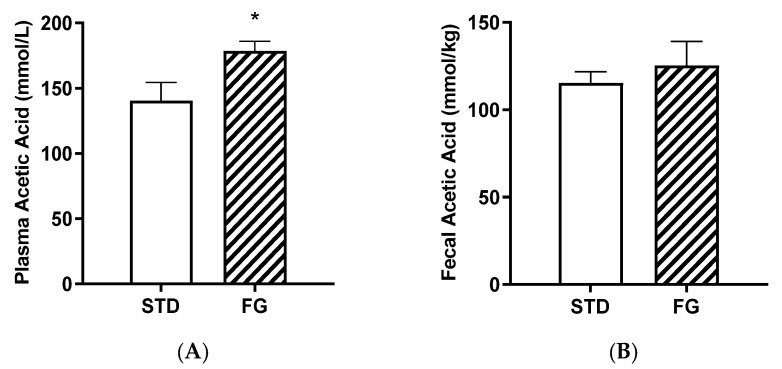
Acetic acid concentrations in plasma (**A**) and feces (**B**) from rats (*n* = 8–9) supplemented or not with d-fagomine for 6 months. Comparisons were conducted using Student’s *t*-test. * *p* < 0.05.

**Figure 5 nutrients-14-04656-f005:**
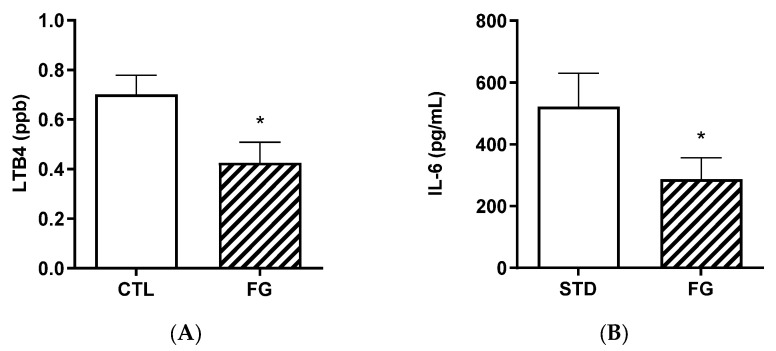
Biomarkers of inflammation, LTB4 (**A**) and IL-6 (**B**) in plasma from rats (*n* = 8–9) supplemented or not with d-fagomine for 6 months. Comparisons were conducted using Student’s *t*-test. * *p* < 0.05.

**Table 1 nutrients-14-04656-t001:** Plasma variables in the STD and FG groups of rats (*n* = 7–9/group) at the end of the study (week 23).

	STD		FG	
	Mean	SEM	Mean	SEM
ALT (U/L)	27	3	33	3
AST (U/L)	70	7	86	5
Insulin (ng/L)	563	116	391	60
Total Cholesterol (nmol/L)	3.61	0.04	3.30	0.03
LDL (nmol/L)	0.43	0.01	0.53	0.02
HDL (nmol/L)	1.15	0.01	1.08	0.01

Statistical significance was evaluated using the Student’s *t* test.

**Table 2 nutrients-14-04656-t002:** β-diversity of the fecal and cecal microbiota in the STD and FG groups of rats (*n* = 8–9/group) at the end of the study (week 23).

		Weight UniFrac	Bray-Curtis	Jaccard
Phylum	STD-FEC vs. FG-FEC	n.s.	n.s.	n.s.
	STD-CC vs. FG-CC	*p*-value (adonis) = 0.041	n.s.	n.s.
Class	STD-FEC vs. FG-FEC	n.s.	n.s.	n.s.
	STD-CC vs. FG-CC	*p*-value (adonis) = 0.031	n.s.	n.s.
Order	STD-FEC vs. FG-FEC	n.s.	n.s.	n.s.
	STD-CC vs. FG-CC	*p*-value (adonis) = 0.032	n.s.	n.s.
Family	STD-FEC vs. FG-FEC	*p*-value (adonis) = 0.002	n.s.	n.s.
	STD-CC vs. FG-CC	*p*-value (adonis) = 0.013	n.s.	n.s.
Genus	STD-FEC vs. FG-FEC	*p*-value (adonis) = 0.001	n.s.	n.s.
	STD-CC vs. FG-CC	*p*-value (adonis) = 0.004	n.s.	n.s.
Species	STD-FEC vs. FG-FEC	*p*-value (adonis) = 0.001	n.s.	*p*-value (adonis) = 0.035
	STD-CC vs. FG-CC	*p*-value (adonis) = 0.005	n.s.	n.s.

Statistical significance was evaluated using the Wilcox test. n.s.— *p*-value > 0.05.

**Table 3 nutrients-14-04656-t003:** Lipid mediators derived from arachidonic acid from rats (*n* = 8–9) supplemented or not with d-fagomine for 6 months.

	STD (ppb)		FG (ppb)	
	Mean	SEM	Mean	SEM
11HETE	6.4	0.8	5.4	0.5
5HETE	22.1	1.7	22.6	1.8
12HETE	61.6	13.2	30.7 *	4.5
15HETE	4.3	0.6	4.8	0.3
20HETE	12.2	0.9	11.3	1.3
TOTAL	106.6	17.2	74.8	8.3

Data are presented as means with their standard error. Comparisons were conducted using Student’s *t*-test. * *p* < 0.05.

## Data Availability

The data presented in this study are available on request from the corresponding author.

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
