# Peer review of "Fiber-like Action of d-Fagomine on the Gut Microbiota and Body Weight of Healthy Rats"

_nutrients, 2022, doi:10.3390/nu14214656_

Round 1

Reviewer 1 Report

Reduction of the α-diversity in fecal samples from the FG group, while increased in cecal samples, need to explain in discussion.

In discussion, diversities of the fecal and cecal microbiota need more detailed explanation, why are they different?

The correlation between body weight, inflammatory factors and microbiota need more discussion.

Author Response

Response to Reviewer 1 Comments

Point 1: Reduction of the α-diversity in fecal samples from the FG group, while increased in cecal samples, need to explain in discussion. In discussion, diversities of the fecal and cecal microbiota need more detailed explanation, why are they different?

 Response 1: We provided a short explanation in the text centered around the increase in the populations of Prevotellaceae. We have now extended the explanation to make the point more clear.

Point 2: The correlation between body weight, inflammatory factors and microbiota need more discussion.

Response 2: This comment made us realize that we had not commented on the relationship between body weight gain and low-grade inflammation in connection with gut microbiota. We have now introduced a sentence at the end of the paragraph devoted to the inflammatory markers.

Reviewer 2 Report

Fiber-like Action of D-Fagomine on the Gut Microbiota and 2 Body Weight of Healthy Rats by a Ramos-Romero S et al., is an interesting manuscript but there are minor concerns that need to be addressed.

The comments are attached below;

1) The impact of chronic administration of D- Fagomine on liver health is unknown at this stage.  Please measure hepatic AST and ALT levels in plasma.

2) Does chronic D- Fagomine administration has got any impact on plasma insulin levels? measure insulin levels in the chronic treatment group.

3) To check the overall systemic inflammation it is ideal to measure a panel including IL-1 beta, TNF alpha, and INF alpha, beta and gamma along with IL-6.

4) Please determine plasma lipid levels including total cholesterol, LDL, and HDL levels in the chronic treatment groups.

5) It is also ideal to measure plasma c-reactive proteins to rule out systemic inflammation.

6) Overall grammar edits and spell checks are required 

Author Response

Response to Reviewer 2 Comments

Fiber-like Action of D-Fagomine on the Gut Microbiota and Body Weight of Healthy Rats by a Ramos-Romero S et al., is an interesting manuscript but there are minor concerns that need to be addressed. The comments are attached below:

Point 1: The impact of chronic administration of D- Fagomine on liver health is unknown at this stage.  Please measure hepatic AST and ALT levels in plasma.

Response 1: We have included these data in the new Table 1, as suggested.

Point 2: Does chronic D- Fagomine administration has got any impact on plasma insulin levels? measure insulin levels in the chronic treatment group

Response 2: D-Fagomine does not have significant impact in plasma insulin of healthy rats. We have included the required data in the new Table 1.

Point 3: To check the overall systemic inflammation it is ideal to measure a panel including IL-1 beta, TNF alpha, and INF alpha, beta and gamma along with IL-6.

Response 3: As we have already include some data about the plasma inflammatory status of the animals (Section 3.6.), we have used our plasma samples in the other analysis required by the reviewer.

Point 4: Please determine plasma lipid levels including total cholesterol, LDL, and HDL levels in the chronic treatment groups.

Response 4: Following the referee recommendation, we have included these data in the new Table 1.

Point 5: It is also ideal to measure plasma c-reactive proteins to rule out systemic inflammation.

Response 5: Similar to point 3, as the general inflammatory status of the animals (Section 3.6.) is already showed, we have used our storaged plasma samples in the other analysis required.

Point 6: Overall grammar edits and spell checks are required.

Response 6: Grammar and spelling have been check.
